

# Oscillations in modified combinants of hadronic multiplicity distributions

**P. Agarwal\*, H. W. Ang, Z. Ong, A. H. Chan and C. H. Oh**

Department of Physics, National University of Singapore

\* p.agarwal@u.nus.edu

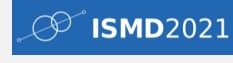

## Abstract

Oscillations in modified combinants ($C_j$s) have been of interest to multiparticle production mechanisms since the 1990s [1]. Recently, there has been a discussion on how these oscillations can be reproduced by compounding a binomial distribution with a negative binomial distribution [2,3]. In this work, we explore a stochastic branching model based on a simple interaction term $\lambda\overline{\psi}\phi\psi$ for partons and propose a hadronization scheme to arrive at the final multiplicity distribution. We study the effects that compounding our model with a binomial distribution has on $C_j$s and explore its physical implications. We find that there is a significant difference in the oscillations in $C_j$s between high energy $pp$ and $p\bar{p}$ scattering that our model can reproduce.

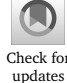

## 1 Introduction

In high energy particle production, one of the characteristic measurements is that of the multiplicity distribution. It assigns a probability $P(n)$ to producing $n$ hadrons for a given interaction. Over the years, many interesting features of these distributions have been studied [4]. Modified combinants can be defined as the coefficients of recursion for the multiplicity distribution. That is

$$(n+1)P(n+1) = \langle n \rangle \sum_{j=0}^{n} C_j P(n-j). \tag{1}$$

These quantities are known to exhibit oscillations w.r.t. their rank $j$. The plots (1) shown showcase these oscillations across three variables: pseudorapidity window, collision type and centre of mass energy. By looking at the plots themselves, there are no obvious candidate

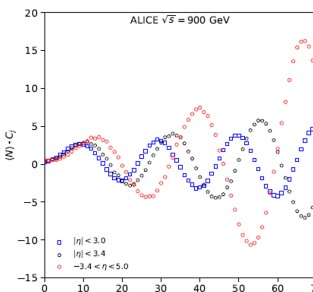
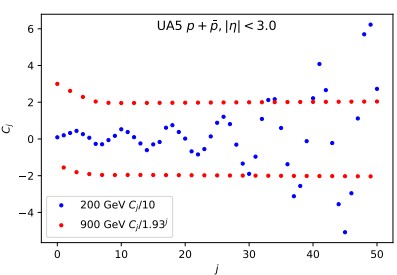

Figure 1: Oscillations in Modified Combinants of hadronic multiplicity distributions with rank.

explanations for what causes these oscillations or what physical information might be gleaned from their period or amplitude.

Herein lies the motivation of this work: we propose a physical model of multiple production that has clear physical meaning to each of its variables. By reproducing these oscillations, we are able to shed some light on what exactly causes this phenomena. We shall focus here mostly on the difference across collision types.

## 2 Multiple Cluster Formation

It was pointed out recently by Wilk *et al.* [2, 5] that clan/cluster models result in oscillations in $C_j$ if the cluster production is explained using a binomial distribution. In such a model, the generating function of the "daughter" distribution that produces particles within a cluster is compounded with the generating function of the "parent" distribution describing cluster production itself. The final multiplicity can be calculated using chain rule on the generating function, as summarised in the Faà di Bruno's formula:

$$P(n) = \frac{1}{n!} \sum_{k=1}^{n} P_{\text{BD}}(r) \cdot B_{n,r}(p_1, 2! \, p_2, ..., (n-r+1)! \, p_{n-r+1}),  \qquad (2)$$

where $B_{n,r}$ are Bell's polynomials and $p_n$ is the daughter distribution.

## 3 Stochastic Branching Models

Hadron production is a QCD phenomena which has inherent non-perturbative effects. One way to circumvent this problem is to employ branching models derived from perturbation theories as the resulting stochastic equations are independent of the coupling [6]. We use one such branching model with the interaction term $\lambda \overline{\psi} \phi \psi$ for our daughter distribution [7]. There are three fundamental processes possible dictated by the same vertex: $\psi$ Bremsstrahlung, $\overline{\psi}$ Bremsstrahlung and $\overline{\psi}\psi$ pair production.

The evolution equation of this system can be easily derived. We assign a probability to each of the three kinds of branching processes: $a$ for $\psi$ Bremsstrahlung, $b$ for $\overline{\psi}$ Bremsstrahlung and $c$ for $\overline{\psi}\psi$ pair production. A state with $n$ "effective" quarks, $m$ "effective" antiquarks and

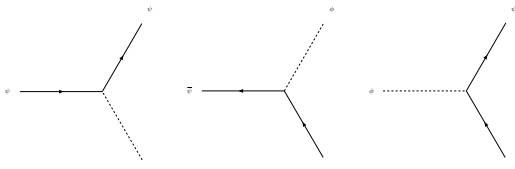

Figure 2: $\psi$ Bremsstrahlung, $\overline{\psi}$ Bremsstrahlung and $\overline{\psi}\psi$ pair production.

$o$ "effective" scalar glueballs has probability given by $P_{mno}$ and evolves as

$$\frac{\mathrm{d}P_{mno}(t)}{\mathrm{d}t} = na(P_{mn,o-1} - P_{mno}) + mb(P_{mn,o-1} - P_{mno})$$
$$+ c((o+1)P_{m-1,n-1,o+1} - o\,P_{mno}). \tag{3}$$

## 4 Hadronisation & Modes of Interaction

### 4.1 Hadronisation

Having generated our partons via branching, we propose a hadronisation scheme that translates this partonic distribution to a final state hadronic distribution. Suppose we have parton multiplicity distributions $p(n)$ for quarks and $\bar{p}(n)$ for antiquarks.

Assumptions:

- We ignore "exotic" particles such as tetra/penta/hexa-quark structures and focus on quark-antiquark mesons and tri quark/antiquark baryons.

- For a given partonic state, all possible physical states have *a priori* the same probability of occurring. For example, if we have 3 quarks and 3 antiquarks, the final state is equally likely to contain either 3 mesons or 1 baryon-antibaryon pair.

Under these assumptions, the hadron multiplicity distribution $P(n)$ is given by:

$$P(n) = \sum_{i=0}^{n}\sum_{j=0}^{n-i} \frac{1}{m_{3(n-i-j)+i,3j+i}}\, p(3(n-i-j)+i)\,\bar{p}(3j+i). \tag{4}$$

Here, $m_{n,\bar{n}}$ are the number of final possible hadronic states for given $n$ quarks and $\bar{n}$ antiquarks. It is symmetric matrix calculated for $n > \bar{n}$ using

$$m_{n,\bar{n}} = 1 + \left\lfloor \frac{\bar{n}}{3} \right\rfloor.$$

### 4.2 Modes of Interaction

We assume a 3-(anti)quark structure for (anti)proton (3). When a $p\bar{p}$ interaction happens, we get a fast moving di(anti)quark and a held-back (anti)quark for each (anti)proton. For hadronisation to result in a final state colour singlet, the held-back quark-antiquark chain hadronise together while the fast moving diquark-diantiquark chain hadronise together [8]. The story is different for $pp$ interactions. The held-back quark pair no longer form a colour singlet and therefore each hadronises with a fast moving diquark instead. This is a *fundamental* difference in the nature of interaction between $pp$ and $p\bar{p}$ that we will be probing.

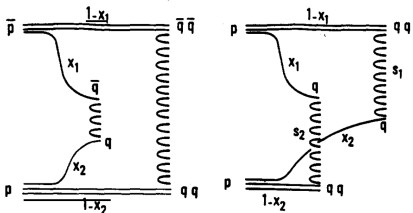

Figure 3: Hadron chains form to keep the final state a colour singlet. (Figure borrowed from [8])

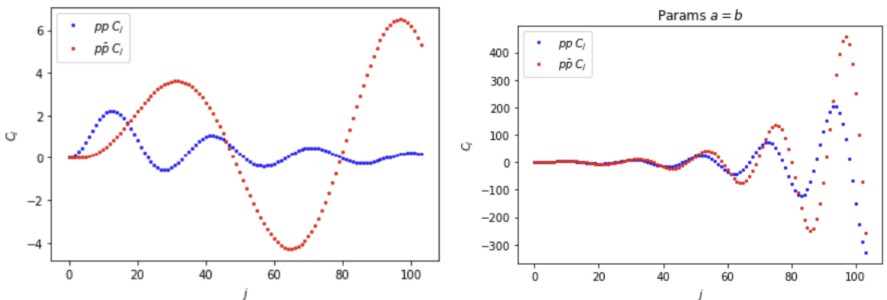

Figure 4: Oscillations in $C_j$ from our model. The plot on the left is for $a \neq b$ while the one on the right is for $a = b$

## 5 Behaviour of $C_j$ in our Model

There are two main ways in our model to distinguish between $pp$ and $p\bar{p}$. First is the difference in branching probabilities of quarks ($a$) and antiquarks ($b$) while the second is the difference between the hadronisation mechanisms of the two interactions. The second of these can be studied simply by constraining the model such that $a = b$. Sample results are shown in Fig. 4.

The effect of $a \neq b$ is brought out by imposing $b \sim 2a$. Notice that the period of oscillation is highly dependent on the branching probabilities of our partons. Another interesting feature of our model is the seemingly linear dependence of the amplitude of oscillations on the maximum number of clusters allowed. The period of oscillation also depends on the probability of cluster formation. These details are currently being investigated.

## 6 Conclusion

The phenomena of oscillations in $C_j$s is an interesting mystery of multiplicity phenomenology. By studying a physical model of these oscillations, there is significant information to be drawn out from the period and amplitude of these oscillations. We find that there is a significant difference between $pp$ and $p\bar{p}$ which our model is able to capture. Upon preliminary fitting, our model is able to describe the multiplicity distribution across $pp$ and $p\bar{p}$ interactions with $\chi^2/d.o.f.$ better than 1.5 in all cases tested (not shown here). The period of oscillation depends on the branching probability of partons as well as the probability of cluster formation.

# Acknowledgements

**Funding information** PA, HWA and ZO are supported by the National University of Singapore Research Scholarship.

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
