# Peer review of "Oscillations in Modified Combinants of Hadronic Multiplicity Distributions"

_SciPost Physics Proceedings, doi:SciPost Phys. Proc. 10, 004 (2022)_

## Round 1 · Referee Report · Anonymous (Referee 1) · 2022-1-3

Strengths

  1. The paper is well-written
  2. The proposed model is novel
  3. The proposed model gives a good qualitative description of Oscillations in Modified Combinants of hadronic multiplicity distributions.

Weaknesses

  1. The paper does not relate the findings of the model to state-of-the-art models for hadron production. For a full paper this may be necessary, but for a conference proceedings the current treatment is fine.

Report

This conference proceedings presents a model based on stochastic branching to explain the difference in oscillations in modified combinants of hadronic multiplicity distributions, for pp and p\bar{p} collisions respectively. By simple means, the authors demonstrate a good qualitative description of data.

For a full paper, I would have suggested the authors to at least:
(1) make a direct comparison between model/data (fig. 1 & 4) to allow more detailed comparison.
(2) compare to state-of-the-art model, such as the Herwig cluster model, which I suspect have a similar behavior, as per the authors' explanation of general features of cluster production of multiplicity. And in case it does not, such a comparison could shed light on some of the more poorly understood features of the colour structure of beam remnants (cf. the authors' sec. 4.2)

For a conference proceedings these remarks are, however, beyond the scope, and I simply leave them here for general inspiration for the authors.

Requested changes

No required changes.

---

## Editorial Decision

published